# Single Nucleotide Polymorphisms in Close Proximity to the *Fibroblast Growth Factor 21 (FGF21)* Gene Found to Be Associated with Sugar Intake in a Swedish Population

**DOI:** 10.3390/nu13113954

**Published:** 2021-11-05

**Authors:** Suzanne Janzi, Esther González-Padilla, Kevin Najafi, Stina Ramne, Emma Ahlqvist, Yan Borné, Emily Sonestedt

**Affiliations:** Department of Clinical Sciences Malmö, Lund University, 20502 Malmö, Sweden; esther.gonzalez_padilla@med.lu.se (E.G.-P.); kevin_najafi@hotmail.com (K.N.); stina.ramne@med.lu.se (S.R.); emma.ahlqvist@med.lu.se (E.A.); yan.borne@med.lu.se (Y.B.); emily.sonestedt@med.lu.se (E.S.)

**Keywords:** genetic variants, SNPs, sugar intake, total sugar, added sugar, sweet taste, *FGF21* gene, *FTO* gene

## Abstract

Hereditary mechanisms are partially responsible for individual differences in sensitivity to and the preference for sweet taste. The primary aim of this study was to examine the associations between 10 genetic variants and the intake of total sugar, added sugar, and sugars with sweet taste (i.e., monosaccharides and sucrose) in a middle-aged Swedish population. Two single nucleotide polymorphisms (SNPs) within the *Fibroblast grow factor 21 (FGF21)* gene, seven top hits from a genome-wide association study (GWAS) on total sugar intake, and one SNP within the fat mass and obesity associated (*FTO*) gene (the only SNP reaching GWAS significance in a previous study), were explored in relation to various forms of sugar intake in 22,794 individuals from the Malmö Diet and Cancer Study, a population-based cohort for which data were collected between 1991–1996. Significant associations (*p* = 6.82 × 10^−7^ − 1.53 × 10^−3^) were observed between three SNPs (rs838145, rs838133, and rs8103840) in close relation to the *FGF21* gene with high Linkage Disequilibrium, and all the studied sugar intakes. For the rs11642841 within the *FTO* gene, associations were found exclusively among participants with a body mass index ≥ 25 (*p* < 5 × 10^−3^). None of the remaining SNPs studied were associated with sugar intake in our cohort. A further GWAS should be conducted to identify novel genetic variants associated with the intake of sugar.

## 1. Introduction

Over the years, scientific evidence associating sugar intake with non-communicable diseases such as dental caries [1,2], weight gain [3,4], metabolic syndrome, type 2 diabetes [1,2,5], and cardiovascular mortality [1] among others, has continued to increase [6,7]. Due to the adverse health effects associated with high sugar intake, it is important to understand the determinants of consumption and desire for foods with a high sugar content.

The study of genetic factors has proved to be an important source of knowledge and understanding of the determinants and mechanisms of dietary preference and consumption through the identification of single-nucleotide polymorphisms (SNPs). The potential SNPs related to sugar intake and sweet taste preference, as well as other traits, can be identified using various approaches. In the candidate gene approach, the studied genes are selected based on their biological function on the phenotypes of interest or their proximity to a chromosomal region that has been linked with the phenotype [8], whereas in genome-wide association studies (GWAS), the whole genome is analyzed without previous assumptions so that new genetic variants may be revealed [9].

Studies using the candidate gene approach have reported associations between sucrose sensitivity and several genetic variants within or in close proximity of the sweet taste receptor genes *TAS1R2* [10], *TAS1R3* [11], and the downstream gene *GNAT3* [12]. However, associations between genetic variants and the intake of carbohydrates and/or sweet foods have only been found for the sweet taste receptor *TAS1R2* [13], but not for *TAS1R3* [14], or for the glucose transporter *GLUT2* [15]. Nevertheless, of the 21 investigated SNPs in these genes, none were found to be associated with total sugar intake in the UK biobank (*n* = 174,424) [16]. In GWAS, the fibroblast growth factor 21 (*FGF21*) gene, more specifically the rs838133 SNP and the adjacent rs838145, have been linked to higher intake of carbohydrates and overall caloric intake [13,17,18,19]. Although this is a well-established region in relation to carbohydrate metabolism and intake, it is understudied regarding specific subgroups of carbohydrate intake, in particular added sugar.

A GWAS study by Hwang et al. [16] aimed to discover new possible genetic variants associated with the intake of total sugar and sweets using data from the UK biobank, perceived intensity of glucose, fructose, and sweeteners using an Australian adolescent twin sample, and perceived intensity and sweetness and the liking of sucrose using a US adult twin sample. The strongest association observed, and the only loci that reached GWAS significance, was between the rs11642841 within the fat mass and obesity associated (*FTO*) gene and total sugar intake (*p* = 3.8 × 10^−8^) [16]. In addition, results found for many of the SNPs and phenotypes were suggestive of an association (*p* < 1.0 × 10^−5^). Therefore, a replication of these associations is required.

The aim of this study was to explore the associations between the eight SNPs most strongly associated with total sugar intake as found by Hwang et al. [16] as well as the two well-established SNPs in close relation to the *FGF21* gene with the consumption of total sugar (i.e., all mono- and disaccharides present in the diet), added sugar (i.e., mono- and disaccharides not naturally occurring in foods and beverages), and sugars with a sweet taste (i.e., all monosaccharides and sucrose) in a middle-aged Swedish population. As a secondary aim, we explored the associations with an intake of different sugar-sweetened foods and beverages, macronutrients, and energy intake. Additionally, we studied 20 SNPs in candidate genes as listed by Hwang et al., and the 71 SNPs identified in their GWAS of sweet intake and perceived intensity of sweet substances [16].

## 2. Materials and Methods

### 2.1. Study Population

The Malmö Diet and Cancer Study (MDCS) is a population-based prospective cohort study, with a baseline examination conducted between years 1991–1996. The source population consisted of men born between 1923 and 1945, and women born between 1923 and 1950, living in the municipality of Malmö, Sweden (*n* = 74,138). All participants provided written informed consent prior to participation, and ethical approval was granted by the Regional Ethical Review Board in Lund (LU/90-51) [20,21].

The participants completed a self-administered questionnaire that included questions regarding lifestyle factors such as education, demographics, alcohol habits, physical activity, tobacco use, and past medical history. Anthropometric measurements were performed, and blood samples were collected from the participants and stored in a biobank. Among the 28,098 individuals with complete anthropometric and dietary information, 27,068 individuals had information about genetics. We excluded 3263 individuals who were born outside of Sweden, as well as 1011 individuals with diabetes at baseline due to a generally reduced intake of sugar in this group. After the exclusions, 22,794 participants remained and constituted the study sample of this study.

### 2.2. Dietary Data

Dietary data were collected through a modified diet history method which consisted of a food diary, a food questionnaire, and a complementary interview. Participants reported their consumption of cooked meals and cold beverages in the food diary for seven consecutive days. The 168-item food questionnaire estimated average frequencies and portion-sizes of food items not covered in the food record (mainly breakfast and snacks) during the preceding year. Additionally, a 60-min (until September 1994) or 45-min (from September 1994) diet history interview was held, where information about the cooking methods and portion sizes was recorded and reviewed so that there was no overlapping information between the food diary and the food questionnaire [22].

The participants’ energy and nutrient intakes were calculated based on information from the Swedish National Food Agency’s database [22]. A validation study has been performed, where the diet history method was validated against 18 days of weighed food records. A relatively high ranking validity was revealed with an energy-adjusted Pearson correlation coefficients of (men/women) carbohydrates (0.66/0.70), protein (0.54/0.53), fat (0.64/0.69), fibre (0.74/0.69) and sucrose (0.60/0.74) [23].

The main dietary outcomes for our study were total sugar intake, added sugar intake and the intake of sugars with a sweet taste. The total sugar intake included all mono- and disaccharides present in the diet from any source. The added sugar intake was estimated by subtracting the naturally occurring sugars in fruit, vegetable, and fruit juice intake from the sum of the participants’ total monosaccharide and sucrose intakes [24]. Sugars with a sweet taste included all monosaccharides and sucrose, both added to and naturally occurring in foods. These variables were expressed as percentages of non-alcoholic energy intake (E%).

The secondary dietary outcomes were monosaccharide intake (mainly fructose, glucose and galactose) (E%), disaccharide intake (mainly lactose, sucrose and maltose) (E%), sucrose intake (E%), sweets and chocolate intake (g/day), sugar-sweetened beverages (SSBs) intake (g/day), ice cream intake (g/day), pastry intake (cakes, pies, cookies, and buns) (g/day), total energy intake (kcal/day), carbohydrate intake (E%), fat intake (E%), and protein intake (E%).

### 2.3. Genotyping and Selection of SNPs

Blood samples were used for genotyping, which was performed using the Illumina GSA v1 genotyping array. Some SNPs were not genotyped directly but were imputed via the Haplotype Reference Consortium reference panel [25].

Our primary exposures consisted of the two SNPs near the *FGF21* gene (rs838133 and rsr838145), and eight SNPs that constituted the top hits for total sugar intake in the UK biobank GWAS (*n* = 174,424) identified by Hwang et al. [16]. Our secondary exposures consisted of 104 SNPs and included 11 SNPs that were suggestively associated (*p* < 1 × 10^−5^) with sweets intake in the UK biobank GWAS (*n* = 21,447) [16] as well as 73 SNPs associated with the perceived intensity and preference of various sweet substances in two samples: the US adult twin sample (*n* = 686) [26] and the Australian Brisbane adolescent Twin Study (*n* = 1757) [27]. Additionally, 20 SNPs that were previously identified using the candidate-gene approach in association with sweet phenotypes and listed in Hwang et al. [16] were also studied (Figure 1).

The genotyped variants were subject to quality control and further exclusions were made in cases of Hardy–Weinberg Equilibrium test of *p* < 1 × 10^−15^ (Appendix A), sample call rate of <90%, and minor allele frequency (MAF) of <0.05. Information about six of the SNPs included in Hwang et al. was unavailable in the MDCS, five SNPs were excluded due to a minor allele frequency (MAF) < 0.05 (Appendix A), and two duplicates were removed, resulting in a total of 101 SNPs that were included in our analyses (Figure 1).

### 2.4. Statistical Analyses

All the statistical analyses were performed using R version 4.0.3 (R Foundation for Statistical Computing, Vienna, Austria). A linear regression model was used to study the associations between the SNPs and the dietary outcomes as continuous variables. The SNPs were coded as 0, 1, and 2, with 2 being the homozygous for the effect allele (i.e., the allele that was reported to be associated with higher outcome ratings in Hwang et al. [16]). The variables for sugar-sweetened foods and beverages were log-transformed as they were not normally distributed. The model was adjusted for age, sex, method (45- or 60-min dietary interviews), and total energy intake (kcal/day). The effect sizes were presented as a β/standard error of the estimate (SEE). A *p*-value of <0.05 denoted statistical significance and Bonferroni-corrected significance thresholds were used to correct for multiple testing. Furthermore, 10 SNPs were included as primary exposures; thus, the Bonferroni-corrected significance threshold was set to *p* < 0.005. The LD and Hardy–Weinberg equilibrium were identified using the Genetics R-package [28]. The power calculations were performed using the genpwr R-package [29].

Several sensitivity analyses were carried out to further explore the associations between the studied exposures and outcomes. To account for the limitations of a single self-reported dietary assessment, a sensitivity analysis was performed which excluded those assessments for which there was an indication that the reported intake may not be representative of long-term intake, i.e., excluding potential energy misreporters and those who had reported drastic dietary changes before a baseline examination (*n* = 14,939). Potential energy misreporters were identified based on the participants’ estimated energy expenditure using Black’s revised Goldberg cut-offs [30].

Some of the SNPs included in our study, particularly those associated with *TAS1R2* [10,31] and *FTO* [32,33], have been previously indicated to be associated with body mass index (BMI) or to have BMI as an effect modifier. Thus, we explored whether any of our associations were dependent on BMI by studying the associations in individuals with BMI < 25 and ≥25 separately. Finally, an analysis that excluded current smokers and those with missing information on smoking was conducted since it has been suggested that smokers might have impaired taste sensitivity (*n* = 16,436) [34,35].

## 3. Results

### 3.1. Study Population Characteristics

Our cohort consisted of 22,794 individuals (61.4% women) with an average age of 58.1 years (ranging between 44.5 and 73.6 years). The study population had a mean energy intake of 2281 kcal per day, a mean total sugar intake of 20.4 E%, a mean added sugar intake of 10.2 E%, and a mean intake of sugars with a sweet taste of 16.0 E%. The mean BMI of our sample was 25.5 kg/m^2^ with 50.8% of participants having a BMI of 25 kg/m^2^ or above. The number of current smokers in our population was 6352 (27.9%) (Table 1).

Information about the 10 primary genetic variants in our analyses is shown in Table 2, including the location, associated gene, effect allele, as well as the results reported for the associations with total sugar intake from Hwang et al. [16].

### 3.2. Associations between Primary SNPs and Main Outcomes

We found several Bonferroni-corrected significant associations (*p* < 0.005) between the primary genetic variants and the three main forms of sugar intake under study (total sugar, added sugar, sugars with sweet taste). The strongest associations were found for the three SNPs located in chromosome 19, within the *FGF21* gene (rs838133) or in close proximity to the *FGF21* gene (rs838145 and rs8103840) (Table 3). All three SNPs were in high linkage disequilibrium (LD), the correlation coefficient between rs838145 and rs838133 was 0.62 (D′ = 0.81) and the correlation coefficients for rs838145 and rs838133 with rs8103840 were 0.68 (D′ = 0.99) and 0.70 (D′ = 0.98), respectively (Appendix A). All three alleles (rs838145 G, rs838133 A, and rs8103840 C) were positively associated with intakes of total sugar (β = 0.18, β = 0.22, and β = 0.20, respectively), added sugar (β = 0.13, β = 0.15, and β = 0.13, respectively), and sugars with a sweet taste (β = 0.16, β = 0.22, and β = 0.20, respectively), with the β-value representing an increase in E% per additional allele (Table 3, Figure 2).

Another significant association was found for the rs60764613 G allele, located within the *CTD-2015H3.1* gene on chromosome 18, which was positively associated with added sugar intake (*p* = 2.89 × 10^−3^) (Table 3 and Appendix A). However, the only SNP with a strong GWAS significance in Hwang et al. [16] was rs11642841 within the *FTO* gene, which did not associate with any of the main sugar outcomes in our cohort.

### 3.3. Associations between Primary SNPs and Secondary Outcomes

Regarding secondary outcomes (sugar-sweetened food and beverage consumption, other forms of sugar intake, energy intake, and macronutrient intakes), rs838145 G allele showed positive associations for intakes of carbohydrates, sucrose, disaccharides, sweets and chocolates, and cakes, but not for SSBs, monosaccharides, ice cream or total energy intakes. The rs838133 A allele, which had a slightly stronger association with the intakes of added sugar and sugars with a sweet taste than the rs838145 G allele, showed similar associations as the former but with the addition of a positive association with monosaccharide intake and a negative association with protein intake. The rs8103840 C allele showed positive associations for intakes of sucrose, disaccharides, and cakes, but not for SSB, ice cream, total energy, or fat intakes (Figure 2 and Appendix A). Further, a significant association was found for the rs60764613, located within the *CTD-2015H3.1* gene for the intake of sweets and chocolate, and cakes. The rs11642841, allocated within the *FTO* gene, presented a significant association with intakes of disaccharides and total energy intake and it was one of few alleles to demonstrate a suggestive association (*p* < 0.05) with intakes of SSBs (Figure 2 and Appendix A).

### 3.4. Associations between Secondary SNPs and All Outcomes

A statistically significant inverse association was identified between the rs5400 C allele (located within the *GLUT2* gene) and carbohydrate intake, and between the rs167132 T allele (previously associated with gSweet, a weighted mean of the glucose, fructose, NHDC, and aspartame intensity ratings, and sucrose intensity traits in the GWAS performed by Hwang et al. [16]) and total energy intake (Figure 2 and Appendix A).

### 3.5. Sensitivity Analyses

When the population was stratified based on BMI, the associations for participants with BMI ≥ 25 (50.8% of the population) were generally stronger than those for BMI < 25. Among participants with a BMI ≥ 25, the associations between rs11642841 within the *FTO* gene (i.e., the only SNP reaching GWAS significance in Hwang et al. [16]) and total sugar, added sugar, and sweet sugars, as well as sucrose intake were strengthened compared to the main results. Indications of interactions between BMI and rs11642841 on total sugar, added sugar, and sugars with sweet taste were found (*p* = 0.04 for all outcomes). For BMI < 25, only a few associations met the Bonferroni-corrected threshold of significance, such as the association of rs838133 with an intake of sugars with a sweet taste and rs60764613 with cake intake (Figure 3, Appendix A).

In further sensitivity analyses, we observed that the exclusion of current smokers strengthened the associations of the three SNPs within or near the *FGF21* gene (rs838145, rs838133 and rs8103840) with total sugar. The associations between rs60764613 and total sugar and sugars with a sweet taste were also strengthened, reaching a significance of *p* < 0.05 but not reaching the Bonferroni-corrected threshold of significance (Appendix A, Appendix A). When potential energy misreporters and diet changers were excluded (34.5% of the population), attenuations of a few associations were observed whereas some associations were strengthened. For instance, rs8103840 (*FUT1* and *FGF21* genes) association with total sugar intake was attenuated but remained significantly Bonferroni-corrected (*p* < 0.005), and the associations of rs838145 (*FGF21* gene) and rs60764613 (*CTD-2015H3* gene) and added sugar intake were attenuated and no longer reached the Bonferroni-corrected significance, whereas the associations between rs838133 (*FGF21* gene) and the main outcomes were strengthened. (Appendix A, Appendix A).

## 4. Discussion

Our study primarily aimed to examine the associations between well-established genetic variants in the *FGF21* gene and different forms of sugar intake, as well as to replicate the top hits recently reported in the GWAS by Hwang et al. [16]. We found significant associations between three previously reported SNPs within and in close proximity to the *FGF21* gene (rs838133, rs838145, and rs8103840) and total intake of sugar, added sugar, and sugars with a sweet taste. In contrast with Hwang et al. [16], no significant associations were found between the rs11642841 within the *FTO* gene in our main analyses. However, when stratifying our sample based on BMI, an association between rs11642841 and the total and added sugar intakes for participants with a BMI ≥ 25 kg/m^2^ was discovered. The remaining SNPs could not be replicated for associations with sugar intake in our cohort, including those within genes coding for proteins involved with the transduction of sweet taste signals, such as the *TAS1R2* and *GNAT3* genes.

Our findings agree with previous GWASs that linked several variants within the *FGF21* locus with macronutrient intake [17,19,36,37], and there is much support for the idea that *FGF21* is the effector gene behind the associations between rs838133, rs838145, and rs8103840 and a higher sugar intake. It has been demonstrated that the liver-derived hormone FGF21, encoded by the *FGF21* gene, is released in response to sugar consumption [13,38], alcohol intake [39] and diets that are deficient in protein [40,41], further contributing to an explanation for the observed associations with a lower protein intake in the present study. This sugar-induced FGF21 response signals for the central nervous system to suppress preference of sweet taste and sugar intake through a negative feedback loop so as to restore macronutrient balance [42,43,44]. This effect has been further demonstrated by the administration of FGF21 analogues in animals [45], and antibody-mediated activation of the FGF21 receptor-complex in humans [46], which both have been found to suppress the sweet taste preference [45,46]. Recent findings in mice have indicated that the primary dietary effect of FGF21 is on sugar and carbohydrate preference, rather than on protein preference per se [47], and effects on protein intake may primarily occur in terms of a substitution for carbohydrates.

When examining whether any of the sugar-sweetened foods or beverages may contribute to associations with sugar intake, connections were found between the three SNPs in close proximity to the *FGF21* gene as well as the rs60764613 (within the *CTD-2015H3* gene) and higher intakes of cakes, and sweets and chocolate. Previously reported findings from MDCS for another SNP within the *FTO* gene (rs9939609) [32], only found associations with cakes and SSB, but no other foods. In our study, suggestive associations were found between the rs11642841 C within the *FTO* gene and the intake of cakes (*p* = 2.7 × 10^−3^) and SSB consumption (*p* = 7.6 × 10^−3^). Furthermore, we did not find associations between any of the other studied SNPs and the intake of SSBs. This was an unexpected finding since SSB consumption tends to be more consistently associated with adverse health outcomes as compared to the intake of total sugar [6].

When dividing the participants based on their BMI, the associations between rs11642841 in the *FTO* gene and most of the associations with sugar intake variables were found exclusively among participants with BMI ≥ 25 kg/m^2^, indicating that these associations might be partially associated with BMI as well. For Hwang et al., the rs11642841 C allele was found to have the strongest positive association with total sugar intake and was, contrary to our findings, inversely associated with BMI [16]. In line with the findings of Hwang et al. [16], but in contrast with previous studies reporting associations between variants within the sweet taste receptor and sweet signal transduction genes *TAS1R2* and *GNAT3*, and sweet perception, sweet preference, and intake of sweet foods [10,31,48,49,50], we found no associations for the variants within *TAS1R2* or *GNAT3* in the main analyses. There were no associations found for *TAS1R2* in the subgroup analyses of BMI < 25 and ≥25, despite the associations between *TAS1R2* and sugar intake previously having been suggested to be BMI dependent [10,48]. These discrepancies could be due to previous studies being conducted in smaller study samples of a few hundred participants and with less comprehensive dietary assessment methods [10,48], and thus, our findings provide further insight to the genetics of sugar intake as related to the sweet taste receptor and sweet signal transduction genes.

When smokers were excluded in a further sensitivity analysis, strengthened associations between the *FGF21* adjacent SNPs and an association between rs60764613 (within the *CTD-2015H3* gene), which was previously associated with smoking initiation [51], and total sugar intake were found. Additionally, the methodology used in this study allowed for the exclusion of potential energy misreporters and drastic dietary changed. Given that the dietary data used in this study was self-reported, this sensitivity analysis aimed to account for potential diet measurement errors and unstable eating habits and resulted in some associations being strengthened, as was the case for the three main SNPs in close proximity with the *FGF21* gene, while other associations were weakened.

There are several possible explanations behind the discrepancies between our results and those reported by Hwang et al. [16]. For example, none of the SNPs, except for rs11642841 within the *FTO* gene, reached the GWAS significance threshold in Hwang et al. [16], thus it is possible that they were chance findings. Another possible explanation is that many of the suggestive associations reported by Hwang et al. [16] were found for perceived intensity and preference of sweet substances, and not for sugar consumption per se, which was the aim of our study. Methodological factors that could influence the discrepancies of our findings include the use of different dietary assessment methods and of populations with different sample sizes. Finally, the fact that the studies were conducted during different time periods could also influence any discrepancies of results found for sugar intake between the studies, as the general consumption patterns could vary through time.

In addition to extensive sensitivity analyses, the strengths of this study include its large study sample and the comprehensive dietary assessment which allowed us to study different dietary factors, including various types of sugars and sources of added sugar, which in turn allowed us to explore whether certain sugars were more strongly associated with the exposures than others. For example, our study found that rs838145 G (*FGF21* gene) was associated with an increased intake of sucrose and other disaccharides, while no association was found with the monosaccharide intake. This study is, to the best of our knowledge, the first to investigate the associations between multiple SNPs and intake of different sugars and sources of sugar. Although we had no information from our participants regarding sweet-taste perception to compare with the results obtained by Hwang et al. [16], being able to measure sugars with sweet taste as an approximation of this information was an additional strength of our study.

Despite the relatively large study sample, it is possible that the study lacked the statistical capacity to identify SNPs with small effect sizes, particularly those with a lower MAF. The dietary outcomes in this study are most likely polygenic traits, meaning that they are influenced by multiple different SNPs with generally low effects. For example, in our study we had a high capacity to detect the effects of the *FGF21* adjacent SNPs, which all had high effect allele frequencies and relatively high effects (β~0.20) for total sugar and sugars with sweet taste, whereas the statistical power may not have been high enough to identify other, less common SNPs with lower effects. In our analysis, we assumed an additive model, but the statistical capacity would be further weakened with regard to dominant or recessive effects. To gain further knowledge about the genetic background of sugar consumption, larger study samples are warranted. Additionally, GWAS should be conducted for the specific outcomes of this study to identify SNPs that are specifically associated with the consumption of various sugars as opposed to a preference or perception of sweetness. Unfortunately, high-quality information was not available regarding the consumption of non-nutritive sweeteners, and 90% of the study population reported that to have no consumption of artificially sweetened beverages. Thus, associations with non-nutritive sweetener intake were not investigated. To gain a deeper understanding of the genetics of sweet preference and consumption, future studies should examine the associations between genetic variants and non-nutritive sweeteners.

Furthermore, it is important to note the potentially limited generalizability of our results due to the homogeneity in ethnicity, locality, and the age of our study population [20], pertaining to both genetic conditions and consumption patterns as the outcomes may reflect diverse behaviors in populations from different countries or age groups. Although there might be some differences between the MDCS population and the samples studied in Hwang et al. [16], they were all of European ancestry and presumably have a comparable genetic architecture. However, these results can not necessarily be extrapolated to populations of other ancestries. Consequently, more studies are needed in European populations to confirm our results, as well as in populations of different ancestries to investigate whether similar associations can be found. Moreover, the dietary data of MDCS was collected in the 1990s, and it therefore reflects the eating patterns of that period in an older adult Swedish population, meaning that different results might be obtained in younger participants or in more recent studies and it should consequently be investigated further.

## 5. Conclusions

This study explored SNPs that have been previously suggested to be associated with sugar intake and sweet taste preference and sensitivity, in association with an intake of numerous different sugar definitions and different sugar-rich foods and beverages in a Swedish population. The strongest associations were found between three variants located within or in close relation to the *FGF21* gene (rs838145, rs838133, and rs8103840) and intakes of added sugar, total sugar, and sugars with a sweet taste, providing additional support for the role of FGF21 in the regulation of sweet taste preference. Most of the previously identified SNPs could not be replicated to associate with sugar intake in this population. These findings contribute important knowledge to the general understanding of genetic determinants of sugar consumption behaviours and provide useful insights for future Mendelian randomization studies that may provide insight into the causality between sugar consumption and disease incidence, which to date remains unclear. Further research should be conducted in populations of different ancestries, age groups, and dietary habits to gain a better understanding of the associations between SNPs and sugar consumption. Additional GWAS should also be conducted to identify novel SNPs that are specific to the different types of sugars investigated in this study.

## Figures and Tables

**Figure 1 nutrients-13-03954-f001:**
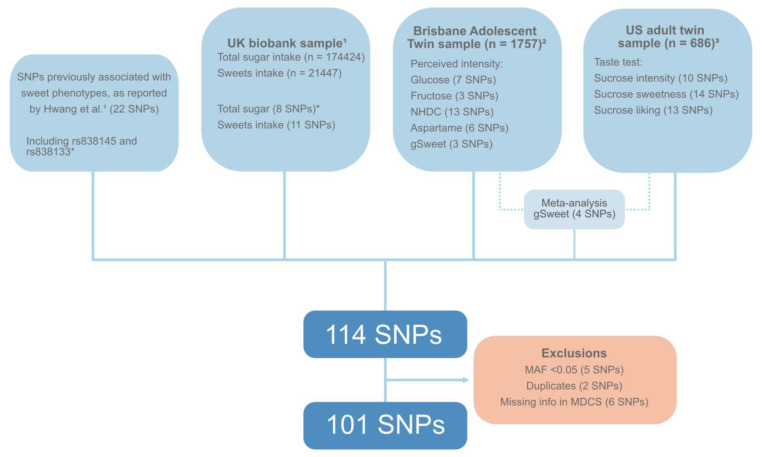
Description of all SNPs included in this study, based on a list compiled by Hwang et al. [16]. * Main SNPs in our study. ^1^ Hwang et al. [16]. ^2^ Hwang et al. [27]. ^3^ Knaapila et al. [26] SNPs: Single-nucleotide polymorphisms. MAF: Minor allele frequency. MDCS: Malmö Diet and Cancer Study. gSweet: Weighted mean of the glucose, fructose, NHDC, and aspartame intensity ratings.

**Figure 2 nutrients-13-03954-f002:**
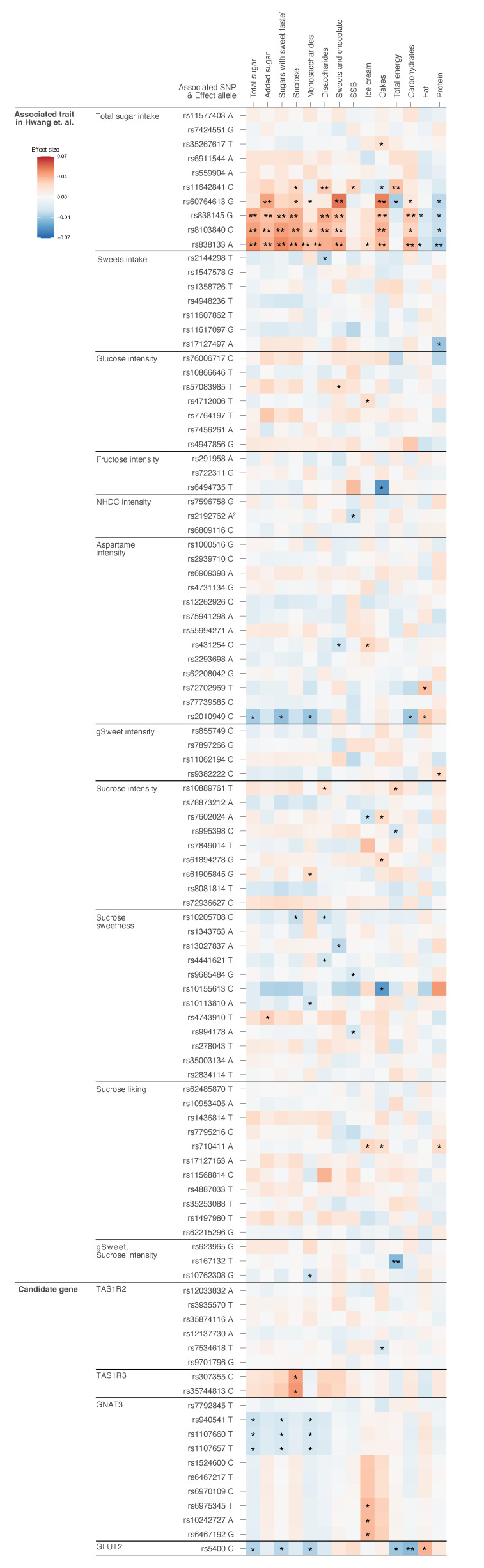
Associations between all primary and secondary SNPs and dietary variables in the MDCS cohort. The effect sizes are presented as β/SEE. TAS1R2: Taste receptor type 1 member 2 gene. TAS1R3: Taste receptor type 1 member 3 gene. GNAT3: G protein subunit alpha transducin 3. GLUT2: Glucose transporter 2 gene. SEE: Standard error of the estimate. MDCS: Malmö Diet and Cancer Study. ^1^ Sucrose and all monosaccharides. ^2^ Rs2192762 was associated with both NHDC and gSweet intensity in Hwang et al. [16]. * *p* < 0.05, ** *p* < 0.005.

**Figure 3 nutrients-13-03954-f003:**
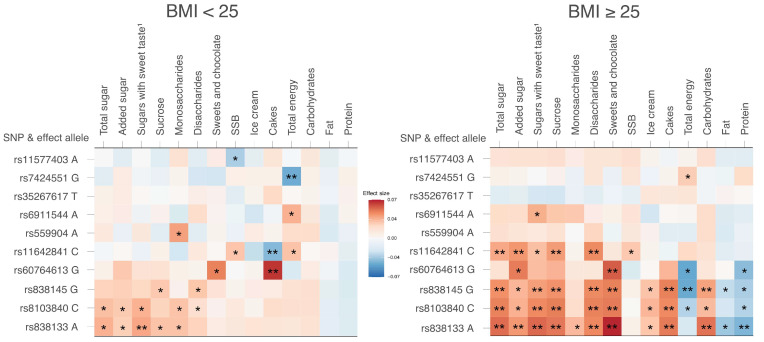
Sensitivity analyses studying participants with BMI <25 and ≥25 separately. The figure shows results for the 10 primary SNPs and all outcomes. The effect sizes are presented as β/SEE. SEE: Standard error of the estimate. ^1^ Sucrose and all monosaccharides. * *p* < 0.05, ** *p* < 0.005.

**Table 1 nutrients-13-03954-t001:** Baseline characteristics for the Malmö Diet and Cancer study population.

Characteristic	Mean (SD)
Age (years)	58.1 (7.68)
BMI (kg/m^2^)	25.5 (3.87)
Total sugar (E%)	20.4 (5.24)
Added sugar (E%)	10.2 (4.21)
Sugars with sweet taste (E%) ^1^	16.0 (4.93)
Sucrose (E%)	8.62 (3.46)
Monosaccharides (E%)	7.36 (2.82)
Disaccharides (E%)	13.0 (3.98)
Sweets and chocolate (g/day)	15.1 (19.9)
Sugar-sweetened beverages (g/day)	74.9 (143)
Ice cream (g/day)	12.1 (18.6)
Cakes (g/day)	38.1 (31.0)
Total energy (kcal/day)	2281 (644)
Carbohydrates (E%)	45.0 (5.97)
Fat (E%)	39.2 (6.06)
Protein (E%)	15.8 (2.50)
	*n* (%)
BMI < 25 ^2^	11,188 (49.1)
BMI ≥ 25 ^2^	11,579 (50.8)
Women	13,992 (61.4)
Smoking status ^3^	
Non-smokers	8754 (38.4)
Former smokers	7682 (33.7)
Current smokers	6352 (27.9)

^1^ Sucrose and all monosaccharides. ^2^ Information on BMI was missing for 27 participants. ^3^ Information on smoking was missing for 6 participants. SD: Standard deviation. E%: Percentage of non-alcoholic energy intake. BMI: Body Mass Index.

**Table 2 nutrients-13-03954-t002:** Description and allele frequencies of the 10 primary SNPs.

SNP	CHR:BP	Associated Gene	EA ^1^	NEA	EAF	*p*-Value for Total Sugar Intake from Hwang et al. ^2^
rs11577403	1:43989773	*PTPRF*	A	G	0.36	1.60 × 10^−^^7^
rs7424551	2:216079163	*AC073284.4*	G	A	0.35	6.70 × 10^−^^8^
rs35267617	5:146693114	*STK32A*	T	C	0.47	3.60 × 10^−^^7^
rs6911544	6:51477640	*RP3-335N17.2*	A	C	0.18	1.00 × 10^−^^6^
rs559904	12:121029354	*POP5*	A	G	0.29	2.90 × 10^−^^7^
rs11642841	16:53845487	*FTO*	C	A	0.41	3.80 × 10^−^^8^
rs60764613	18:1839911	*CTD-2015H3.1*	G	T	0.15	1.20 × 10^−^^7^
rs838145	19:48745473	*IZUMO1*, *FGF**21*	G	A	0.40	2.70 × 10^−^^6^
rs8103840	19:49254955	*FUT1*, *FGF21*	C	T	0.50	5.90 × 10^−^^7^
rs838133	19:49261368	*FGF21*	A	G	0.43	4.80 × 10^−^^7^

CHR: Chromosome, BP: Base pair, EA: Effect allele, NEA: Non-effect allele, EAF: Effect allele frequency. ^1^ The effect alleles were determined based on the effect alleles in Hwang et al. [16]. ^2^ Results from Hwang et al. [16].

**Table 3 nutrients-13-03954-t003:** Associations between the 10 primary SNPs and the main outcomes.

		Total Sugar			Added Sugar			Sugars with Sweet Taste ^1^		
SNP EA	Associated Gene	β	SE	*p*	β	SE	*p*	β	SE	*p*
rs11577403 A	*PTPRF*	0.04	0.05	0.42	0.01	0.04	0.82	0.04	0.05	0.40
rs7424551 G	*AC073284.4*	−0.02	0.05	0.71	0.03	0.04	0.45	−0.01	0.05	0.83
rs35267617 T	*STK32A*	−0.01	0.05	0.76	0.00	0.04	0.90	−0.04	0.05	0.34
rs6911544 A	*RP3-335N17.2*	0.08	0.06	0.22	0.07	0.05	0.20	0.08	0.06	0.19
rs559904 A	*POP5*	0.08	0.05	0.13	0.01	0.04	0.74	0.06	0.05	0.21
rs11642841 C	*FTO*	0.09	0.05	0.07	0.10	0.04	0.01	0.06	0.05	0.17
rs60764613 G	*CTD-2015H3.1*	0.06	0.07	0.33	0.16	0.05	2.89 × 10^−3^	0.12	0.06	0.06
rs838145 G	*IZUMO1*, *FGF21*	0.18	0.05	2.32 × 10^−4^	0.13	0.04	1.53 × 10^−3^	0.16	0.05	6.52 × 10^−4^
rs8103840 C	*FUT1*, *FGF21*	0.20	0.05	2.04 × 10^−5^	0.13	0.04	4.85 × 10^−4^	0.20	0.04	1.06 × 10^−5^
rs838133 A	*FGF21*	0.22	0.05	2.42 × 10^−6^	0.15	0.04	1.87 × 10^−4^	0.22	0.05	6.82 × 10^−7^

^1^ All monosaccharides and sucrose. SNP: Single-nucleotide polymorphism, EA: Effect allele, SE: Standard error.

## Data Availability

The dataset presented in this article are not readily available because of ethical and legal restrictions. Requests to access the dataset should be directed to the Chair of the Steering Committee for the Malmö cohorts, see instructions at https://www.malmo-kohorter.lu.se/malmo-cohorts (last accessed on 3 November 2021).

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
