# Peer review of "Single Nucleotide Polymorphisms in Close Proximity to the Fibroblast Growth Factor 21 (FGF21) Gene Found to Be Associated with Sugar Intake in a Swedish Population"

_nutrients, 2021, doi:10.3390/nu13113954_

Round 1

Reviewer 1 Report

In this study, the authors clarified that two single nucleotide polymorphisms (SNPs) within the FGF21 gene, seven top hits from a GWAS on total sugar intake, and 13 one SNP within the FTO gene were  explored in relation to various forms of sugar intake in 22,794 individuals. These results were consistent with former studies, but there seems to be not new findings. At least, I could not find what the authors newly clarified.

Comments

  1. Did the author check the association between protein feeding and FGF21 SNP?  Reduced protein intake more potently stimulate FGF21 secretion as compared with sucrose feeding.
  2. The authors did not reconfirm the association between most of the previously 423 identified SNPs and sugar intake. Did the author check subgroup analysis (<65yo and >65yo)?
  3. The author should clearly describe what the authors newly identified. 

Author Response

Dear reviewer, Thank you for the valuable comments and suggestions for our manuscript. We have taken your feedback into consideration and adjusted the manuscript accordingly.

  1. Did the author check the association between protein feeding and FGF21 SNP? Reduced protein intake more potently stimulates FGF21 secretion as compared with sucrose feeding.

    The included SNPs and their associations with macronutrient intakes were studied, including protein intake. Our findings for protein were in line with previous studies, as inverse associations between rs838145 G and rs838133 A and protein intake were found. This was previously mentioned in the discussion but has now been updated and further elaborated on (lines 365-379):

    “Our findings are in line with previous GWASs linking several variants within the FGF21 locus with macronutrient intake [17,20,37,38], and there is much support for FGF21 being the effector gene behind the associations between rs838133, rs838145, and rs8103840 and higher sugar intake. It has been demonstrated that the liver-derived hormone FGF21, encoded by the FGF21 gene, is released in response to sugar consumption [19,39], as well as following alcohol intake [40] and diets deficient in protein [41,42], additionally con-tributing to an explanation for the observed associations with lower protein intake in the present study. This FGF21 response signals via the central nervous system to suppress preference of sweet taste and sugar intake through a negative feedback loop to restore macronutrient balance [43-45]. This effect has been further demonstrated by the administration of FGF21 analogues in both animals and human, which has shown to suppress sweet taste preference [46,47]. Recent findings in mice have indicated that the primary dietary effect of FGF21 is rather on sugar and carbohydrate preference, than on protein preference per se [48], and effects on protein intake may primarily be in terms of substitution for carbohydrate.”

  1. The authors did not reconfirm the association between most of the previously 423 identified SNPs and sugar intake. Did the author check subgroup analysis (<65yo and >65yo)?

We did not carry out subgroup analysis for <65 years and >65 years, but based on the existing literature, subgroup analysis was carried out for BMI <25 and ≥25 as well as based on smoking status. Several posible explanations for the discrepancies in results between our and previous studies have been mentioned in lines 476-487:

"There are several possible explanations behind the discrepancies between our results and those reported by Hwang et al [16]. For example, none of the SNPs, except for rs11642841 within the FTO gene, reached the GWAS significance threshold in Hwang et al. [16], thus it is possible that they were chance findings. Another possible explanation is that many of the suggestive associations reported by Hwang et al. [16] were found for perceived intensity and preference of sweet substances, and not for sugar consumption per se, which was the aim of our study. Methodological factors that could influence the discrepancies in findings include the use of different dietary assessment methods and in populations with different sample sizes. Finally, the fact that the studies were conducted during different time periods could also influence any discrepancies of results found for sugar intakes between the studies, as the general consumption patterns could vary through time."

  1. The author should clearly describe what the authors newly identified. 

    Following this comment, we have clarified our findings and their relevance to this field of research (line 534-541):

    “This study explored SNPs previously suggested to associate with sugar intake and sweet taste preference and sensitivity, in association with intake of numerous different sugar definitions and different sugar-rich foods and beverages in a Swedish population. The strongest associations were found between three variants located within or in close relation to the FGF21 gene (rs838145, rs838133, and rs8103840) and intakes of added sugar, total sugar, and sugars with sweet taste, providing additional support for the role of FGF21 in regulation of sweet taste preference. Most of the previously identified SNPs could not be replicated to associate with sugar intake in this population.”

Reviewer 2 Report

In this study, Janzi S. examine the association between genetics variants, including variants in FGF21 gene, previously reported by Hwang et al, 2019, and different form of sugar intake (total sugar, added sugar and sugars with sweet taste) using data collection from the Malmö Diet and Cancer Study (MDCS) a middle-aged Swedish population cohort study. They also study the associations with intake of different sugar-sweetened foods and beverages, macronutrients, and energy intake.

This is an interesting approach to try to replicate previous associations between SNPs and sugar intake. The introduction give relevant definition of the different type of study of genetics factors, discussion is well written and addresses the various points that may have biased the differences obtained in other studies. I suggest considering the manuscript for publications after the authors have clarified certain points.

1) High potency sweeteners, like saccharin, aspartame, cyclamate, were already used in food industry the 1990s, why were they not included in the present study ? If these data are recoverable and could be added, it would greatly increase the impact of the analysis since sweeteners are currently widely used but little is available in this kind of analysis. If it is impossible, a comment could be added in the manuscript.

2) As UK BioBank concerned samples collected from 2006-2010 and MDCs was conducted between years 1991-1996, is it possible that this time difference (10 and almost 20 years) lead to change in food habits, leading in general to more impact on obesity that could explain the difference in results.

3) Line 43-45: much more studies than only Dias et al, 2015, Habberstad et al, 2017 and Fushan et al, 2009, have reported impact of SNPs of TAS1R2 or TAS1R3 on dietary intake and/or taste preferences. Some of these studies were published recently (for exemple: Chamoun, 2021, Choi, 2021 …) and should be added and commented on.

4) The link between number cited in paragraph “2.3 Genotyping and selection of SNPs” and Figure 1 is difficult to follow. For example, line 132 identify 20 SNPs from Hwang et al, but the Figure 1 show 22 SNPs in the left square. Line 129 it is write “73 SNPs associated with…”, but in Figure 1, right square for US adult Twin, the addition of all SNPs is 37. Text and/or figure must be clarified.

5) The line 141 repeat the line 137. Line 272 write two consecutive “in”. For table 2, the order of  SNPs in first column should be the same as for Table 3 and Figure 2, in order to facilitate reading and comparison.

Author Response

Dear reviewer, Thank you for the valuable comments and suggestions for our manuscript. We have taken your feedback into consideration and adjusted the manuscript accordingly.

1) High potency sweeteners, like saccharin, aspartame, cyclamate, were already used in food industry the 1990s, why were they not included in the present study? If these data are recoverable and could be added, it would greatly increase the impact of the analysis since sweeteners are currently widely used but little is available in this kind of analysis. If it is impossible, a comment could be added in the manuscript.

Information about specific non-nutritive sweeteners were not available for this population, and only a small proportion of the study population reported to consume artificially sweetened beverages (n=2,194 (9.6%)), which has now been mentioned in the manuscript (line 510-515). This shortcoming of this study is likely a reflection of the time of the study covered, and therefore newer studies could be more appropriate for looking at the associations between the genetic variants and consumption of high potency sweeteners.

“Unfortunately, no high-quality information was available about consumption of non-nutritive sweeteners, and 90% of the study population reported no consumption of artificially sweetened beverages. Thus, the associations with non-nutritive sweetener intake were not investigated. To gain a deeper understanding of the genetics of sweet preference and consumption, future studies should examine the associations between genetic variants and non-nutritive sweeteners.”

2) As UK BioBank concerned samples collected from 2006-2010 and MDCs was conducted between years 1991-1996, is it possible that this time difference (10 and almost 20 years) lead to change in food habits, leading in general to more impact on obesity that could explain the difference in results.

The time of the studies can definitely help explain the difference of our results and the results from the UK biobank. We did mention the influence that the different dietary patterns during different time periods could have on the results (line 525-531): "Additionally, the dietary data of MDCS was collected in the 1990s, therefore it reflects the eating patterns of that time in an older adult Swedish population, and thus, different results might be obtained in younger participants or in more recent studies and should consequently be investigated further".

We have clarified in the discussion that this could also contribute to the discrepancies between our results and those from the UK biobank (lines 484-487): "Finally, the fact that the studies were conducted during different time periods could also influence any discrepancies of results found for sugar intakes between the studies, as the general consumption patterns could vary through time."

3) Line 43-45: much more studies than only Dias et al, 2015, Habberstad et al, 2017 and Fushan et al, 2009, have reported impact of SNPs of TAS1R2 or TAS1R3 on dietary intake and/or taste preferences. Some of these studies were published recently (for exemple: Chamoun, 2021, Choi, 2021 …) and should be added and commented on.

Thank you for suggesting these recent papers, they have been added to this paper and have been commented on in relation to our findings (line 396-400):

“In line with the findings by Hwang et al. [16], but in contrast with previous studies that reported associations between variants within the sweet taste receptor gene TAS1R2 and sweet preference and intake of sweet foods [10,32,49-51], we found no associations for variants within TAS1R2 in the main analyses or in subgroup analyses of BMI <25 and ≥25”

4) The link between number cited in paragraph “2.3 Genotyping and selection of SNPs” and Figure 1 is difficult to follow. For example, line 132 identify 20 SNPs from Hwang et al, but the Figure 1 show 22 SNPs in the left square. Line 129 it is write “73 SNPs associated with…”, but in Figure 1, right square for US adult Twin, the addition of all SNPs is 37. Text and/or figure must be clarified.

  • The flow chart and text have been updated to clarify the SNP selection. The 22 SNPs in the left square refer to rs838145 and rs838133 (main SNPs) as well as 20 additional SNPs as secondary SNPs which has been clarified in the flow chart.

  • On line 129, the 73 SNPs refer to those in both the US adult twin sample and the Brisbane Adolescent Twin sample, which has now been further clarified (132-137):

    “Our secondary exposures consisted of 104 SNPs and included 11 SNPs that were suggestively associated (P < 1 × 10-5) with sweets intake in the UK biobank GWAS (N = 21,447) [16] as well as 73 SNPs associated with perceived intensity and preference of various sweet substances in two samples: the US adult twin sample (N = 686) [27] and the Australian Brisbane adolescent Twin Study (N = 1,757) [28].”

5) The line 141 repeat the line 137. Line 272 write two consecutive “in”. For table 2, the order of  SNPs in first column should be the same as for Table 3 and Figure 2, in order to facilitate reading and comparison.  

In order to clarify the manuscript based on comment 5), the following updates have been made:

  • Line 140-158 has been rewritten as not to repeat the same information twice:
    “The genotyped variants were subject for quality control and further exclusions were made in cases of Hardy-Weinberg Equilibrium test of P < 1 × 10-15 (Supplemental table 1), sample call rate <90%, and minor allele frequency (MAF) < 0.05. Information about six of the SNPs included in Hwang et al. was unavailable in the MDCS, five SNPs were excluded due to a minor allele frequency (MAF) <0.05 (Supplemental table 2), and two duplicates were removed, resulting in a total of 101 SNPs included in our analyses (Figure 1).”

  • One of the two "in" on line 272 (now line 307) has been removed.

  • Table 3, figure 2, and figure 3, as well as the supplementary materials have been updated with the SNPs being in the same order as in table 2.

Round 2

Reviewer 1 Report

The authors did not response to my comments. Many readers including me want to know whether your study provides us new findings in this study. your study only showed the confirmation that FGF21 SNP are associated with sweet intake in a swedish population. This may be an important finding in the aspect of genetic research, but I think that this study has little contribution in nutrition research.  The author should describe the points that this study could contribute to nutrition study.

Author Response

Dear reviewer, we agree that it is very important that the findings and contributions to the research field are clear to the reader. Thank you for the feedback and opportunity to clarify the findings.

A key factor in understanding nutrition is identifying the determinants of dietary intakes, and one such determinant is genetics. Various different studies have identified SNPs associated with carbohydrate preference and intake. Recently, Hwang et al. also identified SNPs associated with total sugar intake and intake of sweets in the UK biobank. There is still a gap in the understanding of genetic determinants of sugar intake, and our study is an important contribution to this research area by:

1. Replicating the previous findings of the GWAS by Hwang et al. about sugar intake in a Swedish cohort with total sugar intake, as well as subgroups of sugar intake and sugar-rich foods and beverages. As health outcomes have been indicated to vary depending on type and source of sugar, it is naturally of interest to decipher the genetics of consumption of different types of sugar and sources of sugar, and this has to the best of our knowledge not been investigated previously. This has now been clarified in the manuscript (line 494-502):

“… the strengths of this study include the large study sample and comprehensive dietary assessment which allowed us to study different dietary factors, including various types of sugars and sources of added sugar, which in turn allowed us to explore whether certain sugars were more strongly associated with the exposures than others. For example, our study found that rs838145 G (FGF21 gene) was associated with increased intake of sucrose and other disaccharides, while no association was found with monosaccharide intake. This study is, to the best of our knowledge, the first one to investigate the associations between multiple SNPs and intake of different sugars and sources of sugar.

2. Investigating if the SNPs previously associated with carbohydrate intake and/or preference are associated with sugar intake in our cohort. Many of the previous findings have been made in small samples or in studies with less comprehensive dietary data (thus not being able to study different types of sugar intakes). For example, like Hwang et al., we were unable to replicate associations for TAS1R2 and GNAT3 despite them previously being consistently associated with sweet preference and/or intake in smaller samples (a few hundred participants). In light of this, our study is valuable with its large study sample and comprehensive dietary assessment, and the fact that only a few SNPs could be replicated with these outcomes in this cohort is noteworthy and an important finding. This has now been further discussed in the discussion (line 398-472):

“In line with the findings by Hwang et al. [16], but in contrast with previous studies reporting associations between variants within the sweet taste receptor and sweet signal transduction genes TAS1R2 and GNAT3, and sweet perception, sweet preference and intake of sweet foods [10,32,49-51], we found no associations for variants within TAS1R2 or GNAT3in the main analyses. No associations were found for TAS1R2 in subgroup analyses of BMI <25 and ≥25, despite the associations between TAS1R2 and sugar intake previously having been suggested to be BMI dependent [10,49]. These discrepancies could be due to previous studies being conducted in smaller study samples of a few hundred participants and with less comprehensive dietary assessment methods [10,49], and thus, our findings provide further insight to the genetics of sugar intake related to sweet taste receptor and sweet taste signal transduction genes.”

Finally, we clarified the findings and their relevance to the field of nutrition further in the conclusion (line 557-561):

These findings add important knowledge to the general understanding of genetic determinants of sugar consumption behaviours and provide useful insights for future Mendelian randomization studies that may answer the question of causality between sugar consumption and disease incidence, which to date remains unclear

Finally, we thank you for helping us improve the quality of this paper, and hope you agree that these findings are important contributions to the field of nutritional genomics.